# Psychosis risk among pregnant women in Ghana

**Samuel Adjorlolo** [1,2]*, **Gwendolyn Mensah** [3], **Caroline Dinam Badzi** [4]

**1** Department of Mental Health Nursing, School of Nursing and Midwifery, College of Health Sciences, University of Ghana, Accra, Ghana, **2** Research and Grant Institute of Ghana, Accra, Ghana, **3** Department of Adult Health Nursing, School of Nursing and Midwifery, College of Health Sciences, University of Ghana, Accra, Ghana, **4** Department of Maternal and Child Health Nursing, School of Nursing and Midwifery, College of Health Sciences, University of Ghana, Accra, Ghana

\* sadjorlolo@ug.edu.gh

## Abstract

### Introduction

Psychotic illness, although is rare, has been reported in the perinatal period. Individuals diagnosed with psychotic illness tend to first exhibit psychotic-like experiences (PLEs), defined as subclinical psychotic symptoms that occur outside the context of sleep or drug use. However, there is a paucity of empirical data on PLEs in pregnancy to advance scholarly discourse and support professional practice. The current study investigated the prevalence and correlates of PLEs among pregnant women in Ghana, a West African state.

### Design

A cross-sectional survey design was used to collect data from 702 pregnant women who responded to measures of PLEs, COVID-19 concerns and behavioral maladies such as anxiety and depressive symptoms. Descriptive and inferential statistics, namely chi square, exploratory factor analysis, MANOVA and multinomial logistic regression were used to analyze the data.

### Results

The results showed that 54.2%, 27.3% and 18.5% of participants were at no/low, moderate and high risk for psychosis, respectively. A total of 44.4% participants were not distressed by PLEs, whereas 32.2% and 23.4% were a bit/quite and very distressed, respectively. Psychosis risk was elevated among pregnant women who were more concerned about the COVID-19 effects, scored high in suicidal ideation, depressive symptoms and sleep difficulties.

### Conclusion

The study showed that psychosis risk is present in pregnancy.

**Data Availability Statement:** All relevant data are within the paper and supporting information file.

**Funding:** The project received financial support awarded to Samuel Adjorlolo from the Canadian Queen Elizabeth II Diamond Jubilee Advanced Scholars Program (QES-AS). The QES-AS is made

possible with financial support from IDRC and SSHRC. Website of funding support: https://www.univcan.ca/programs-and-scholarships/queen-elizabeth-scholars/qes-advanced-scholars/ The funders had no role in study design, data collection and analysis, decision to publish, or preparation of the manuscript.

**Competing interests:** The authors have declared that no competing interests exist.

## Implications

Screening for psychosis risk in pregnancy should be prioritized for pregnant women with behavioral maladies, including suicidal tendencies, depressive symptoms, sleep difficulties and heightened concerns about COVID-19.

## Introduction

Psychotic illness is a rare mental health condition that has been reported among women in the perinatal period [1]. In the maternal mental health literature, limited attention has been granted to psychotic illness relative to anxiety and depression [2, 3]. The available data, nonetheless, suggest quite disturbing prevalence rates and burden of psychosis in the perinatal period. An earlier study found that approximately 59% of mothers recruited from mental health settings were diagnosed with psychosis [4]. In the United States, it has been estimated that about 700 out of 100,000 post-delivery women were hospitalized because of psychotic illness [5]. A study conducted among 745,596 first-time mothers in Sweden revealed that, 892 were hospitalized for psychotic illness and 436 had not previously been hospitalized for any psychiatric disorder [1]. Data on the prevalence of psychosis in pregnancy is limited in Africa owing to lack of studies. Previous studies addressing this topical issue have focused on the factors contributing to the development of mental health issues, including psychosis, post-delivery [6].

Psychotic illness is a major risk factor for death by suicide in childbearing women, with estimates suggesting that one in every 500 women with postpartum psychosis die from suicide [7]. The risk for other mental health problems such as bipolar and depressive disorders is highly elevated in women with a history of postpartum psychotic illness [8, 9]. Psychotic illness has been associated with adverse obstetric and neonatal outcomes such as antepartum hemorrhage, placental abruption, postpartum hemorrhage, premature delivery, stillbirth, premature rupture of membranes, fetal morbidities and mortalities [5, 10]. As a heterogeneous mental disorder, several risk factors have been documented, including childhood maltreatment [11], low socio-economic status, neighborhood level social deprivation [12]. Among the perinatal factors implicated in psychosis include diabetes in pregnancy, antepartum haemorrhage, pre-eclampsia, maternal stress during pregnancy, pre-pregnancy and pregnancy obesity and cord complications [12].

More importantly, studies have suggested that the psychotic pathway commences with what is referred to as psychotic-like experiences (PLEs) [13]. PLEs are subclinical symptoms of psychosis that do not meet the threshold for clinical diagnosis as psychotic illness [13, 14]. PLEs are very common in the general population, appearing first in adolescence and sometimes in childhood [15] and can be categorized as positive (e.g., perceptual abnormalities, delusional thoughts) or negative (e.g., social withdrawal, avolition) [16]. Among the common examples of the PLEs are hearing voices, seeing things, and smelling things that other people do not hear, see or smell, respectively. PLEs are unrelated to or occur outside the context of sleep or drug use [14]. Just like psychotic illness, PLEs have been associated with a decline in general health status and behavioral maladies such as suicidal tendencies, emotional disorders and illicit substance use [17, 18]. These behavioral problems can contribute to mortality and morbidity during and after pregnancy. Because some people with PLEs go on to develop psychotic disorders [13], there is the likelihood that some women diagnosed with postpartum psychosis exhibited PLEs during pregnancy and/or after delivery.

While the foregoing suggests that studies elucidating PLEs in pregnancy are extremely important, our understanding of psychosis risk in pregnancy is limited largely because the

existing studies have overwhelmingly focused on persons who have been diagnosed with psychotic disorders [5, 8–10]. For example, in their study involving 40 mothers diagnosed postpartum psychosis in South Africa, Voges et al. found that substance use during pregnancy, postpartum abuse and lifetime experience of trauma were reported by the participants [6]. Understanding the prevalence and correlates of PLEs in the perinatal period can strategically support healthcare professionals in their effort to render basic mental health services to pregnant women. It is posited that screening for PLEs will enable health professionals to engage with pregnant women on their mental well-being such that those who score high on PLEs could be deemed as high risk for future psychosis. The work by Levey et al. involving Peruvian pregnant women appeared to be the first study to have focused on PLEs in the pregnancy period [7]. Levey et al. reported that 27% of the 2,059 pregnant women scored high on psychosis risk. As noted previously, studies dedicated to PLEs in pregnant women are extremely important. Thus, building on Levey et al.'s study, the current study investigated PLEs among pregnant women in Ghana. The study objectives were to investigate the prevalence of PLEs in pregnant women and secondly determine the correlates of PLEs.

## Methods

### Data source

Data were gathered from 702 pregnant women recruited from the antenatal clinics of four health facilities, namely University of Ghana hospital ($n = 175$, 24.9%), Alpha hospital ($n = 239$, 34%), Sanford Clinic ($n = 87$, 12.4) in the Greater Accra and St-Gregory Catholic hospital ($n = 201$, 28.6%) in the Central regions of Ghana. These facilities are patronized by pregnant women of different demographic backgrounds.

### Study design and data collection

A cross-sectional design was used. The inclusion criteria for participating in the study were age ≥18 years, nulliparity/multiparity and completion of senior high school diploma or equivalent (i.e., 12 years of formal education). This requirement assured the participants were proficient in English. It was most practical to use English questionnaires as there are more than 10 languages spoken in the Greater Accra and Central region of Ghana. Pregnant women in their first (≤ 12 weeks of pregnancy), second (≤ 24 weeks of pregnancy) or third trimesters (≤ 36 weeks of pregnancy) were recruited for the study. The exclusion criteria were diagnosis of a mental disorder during pregnancy and/or a history of mental health disorder. Data were collected by research assistants (RAs). At each facility, a local nurse or midwife was identified as facility-based focal persons. These individuals supported the recruitment process by introducing the research assistants to the participants attending the antenatal clinics. Thereafter, the RAs approached and discussed the study with individual pregnant woman. The participants, who often congregate at the antenatal clinics of the facilities for their antenatal services, were informed about the purpose and duration of the study, their responsibilities for participating in the study, ethical issues such as confidentiality, consent, anonymity and benefits of participation. Questions raised by the participants were responded to by the research team to allay fears, anxieties to encourage participation in the study.

Prior to completing the questionnaire, the participants read and signed the consent form, together with the RAs. The "broad consent" gave the research team the permission to obtain data on pregnancy outcomes and other pertinent obstetric information after delivery from the participants' folders, where necessary. The folder/hospital identity numbers of the participants were recorded to facilitate subsequent matching of information. The questionnaires were completed individually and independently. The RAs were present to provide the needed support to

the participants. Once completed, the questionnaires were handed over to the RAs. Data were collected from September 2020 to October 2020. The COVID-19 precautionary measures such as wearing of nose mask and use of alcohol-based hand sanitizers were strictly adhered to. The study received ethics clearance from the Noguchi Memorial Institute for Medical Research, University of Ghana (NMIMR-IRB CPN 057/19-20).

## Data collection measures

*Prodromal Questionnaire* (PQ-16) was used to measure attenuated symptoms of psychosis or PLEs [19]. The PQ-16 was developed from the 92-item Prodromal Questionnaire (PQ-92) as a brief screening measure. As a self-report questionnaire, the PQ-16 screen for PLEs on a two-point scale (true/false). The PQ-16 raw scores indicate the number of PLEs a participant endorsed. The raw scores are obtained by totaling the number of true responses. The PQ-16 demonstrated good psychometric properties among patients seeking mental health care [19] and pregnant women [7].

*Patient Health Questionnaire-9* (PHQ-9) is a 9-item self-report questionnaire administered to assess for depressive symptoms among the participants [20]. The PHQ-9 items are rated on a four-point Likert scale ranging from 'not at all' (0) to 'nearly every day' (3). Higher scores indicate more depressive symptoms. The internal consistency of the PHQ-9, indexed by Cronbach's alpha, in this sample was 0.79.

*Generalized Anxiety Disorder* [21] scale is a 7-item scale administered to assess the symptoms of anxiety in the participants. The GAD-7 items are rated on a 4-point Likert scale, ranging from 0 (not at all) to 3 (nearly always). Total score on the GAD-7 is obtained by summing the individual items, with high scores indicating more symptoms of generalized anxiety. The Cronbach alpha of the GAD-7 was 0.85.

**COVID-19 concerns.** We assessed COVID-19 concerns using two items that were scored on a four-point Likert response scale from Not at all (0) to Very often (3). The first item relates to whether the participants were worried about contracting the virus and the second involved whether the participants were worried that their babies could develop some birth or developmental abnormalities should they contract the coronavirus. A total score was obtained by summing the responses, with higher scores indicating more COVID-19 concerns. A Cronbach's Alpha of 0.92 was obtained for the two-item "COVID-19 concern" scale.

**Sleep difficulty.** This was measured by asking about (1) difficulty to fall asleep while in bed and (2) difficulty to stay asleep through the night. The items were extracted from the existing literature [22] and were scored using a four-point Likert response format ranging from Not at all (0) to Very often (3). Responses to each item were added to create a total score, with higher scores indicating more sleep difficulty. The Cronbach's Alpha for the sleep difficulty scale was 0.86.

**Suicidal ideation.** Following a review of the literature [23], three items were extracted to index suicidal ideation as follows: Have you (1) ever thought that life wasn't worth living; (2) ever thought about killing yourself; and (3) ever attempted to kill yourself? The response options ranged from Never (0) to Yes, several times (3).Total scores, obtained by summing the responses on the scale, ranged from 0 to 9. Higher scores reflect more suicidal ideation. The Cronbach's Alpha for the suicidal ideation scale was 0.71.

## Other factors

**Intimate partner violence.** This was operationalized using two items from the literature [24, 25]: (1) I have been belittled by my partner (i.e., abuse) and (2) I experienced no attention from my partner (i.e., neglect).

**Help-seeking for mental health.**   Self-initiated help-seeking was measured with a single item: I have sought help for emotional or mental health issues while pregnant in the last 3 months. Based on the recommendations from the NICE guidelines [26], the participants responded to two items to estimate the extent to which health professionals were involved in promoting their mental well-being: (1) Nurses and midwives have asked me about my mental health or emotional well-being and (2) Nurses or midwives have offered me support such as counselling and referral for my mental health or emotional well-being needs. The response options for self-initiated help-seeking and health professional involvement were as follows: Never; Yes, once; Yes, twice and Yes, several times. These were subsequently recoded into two categories for each item: "Never" (coded 0) and "at least once" (coded 1).

## Data analysis

To determine the clustering of the items on the psychosis-risk measure, we fitted principal component analysis (PCA). In addition to eigenvalue and Cattel's scree plot criteria, the decision on the number of components to retain was also based on the results of the parallel analysis and Velicer's Minimum Average Partial (MAP) tests [27]. Because there is no cut-off point on the PLEs measure to determine psychosis risk among pregnant women, we categorized the participants into three groups based on their PLEs scores. First, we converted the psychosis risk scores into standard (i.e., z) scores with a mean of zero and standard deviation of one. Second, scores that were 1 standard deviation below the mean were designated as no/low risk group; scores1 above the mean as high risk group and scores in between as moderate risk group. Descriptive statistics was used to determine the percentage of participants in each psychosis-risk group.

The relationship between psychosis risk group and the categorical study variables (i.e., level of education, pregnancy trimester, partner abuse, mental health help-seeking) was analyzed with chi-square ($\chi^2$). Next, a one-way multivariate analysis of variance (MANOVA) was used to determine whether the psychosis-risk groups differ significantly on the continuous study variables. A Bonferroni-adjusted univariate analysis of variance (ANOVA), with significance level at 0.01(.05/5) was used as a follow-up on the significant MANOVA results. Effect sizes were estimated with partial eta squared ($\eta^2$). Prior to the MANOVA, we performed zero-order correlations using Pearson correlation coefficient to determine the correlations between the continuous variables.

Lastly, a standard multinomial logistic regression was used to predict psychosis risk group membership of the participants. The no/low risk group was used as the reference category against which the moderate and high risk groups were compared. The predictor variables were COVID-19 concerns, depressive symptoms, anxiety symptoms, suicidal ideation and sleep difficulty. The predictor variables in the multinomial logistic regression equation were standardized to mean 0, standard deviation 1 to facilitate the interpretation of the results. The data analyses were performed using SPSS Version 23 (IBM.corp) and an alpha level of 0.05, unless indicated otherwise.

## Results

### Demographic characteristics of participants

**Participants.**   The participants were recruited from the various pregnancy periods (1st trimester = 63, 9%; 2nd trimester = 315, 44.9% and 3rd trimester = 324, 46.2%). More than half (*n* = 358, 53%) completed senior high school/equivalent (at least 12 years of education), 168 (24.9%) completed post-secondary school (additional 2 or 3 years of schooling from senior

high school) whereas 22.1% ($n$ = 149) completed university education. The average age of the participants was 30 years ($SD$ = 5.54).

## Principal component analysis

PCA was conducted to investigate the clustering of the psychosis-risk items. The Kaiser–Meyer–Olkin measure of 0.85 and Bartlett's test of sphericity, $\chi^2$ (120) = 2750.76, p < .001 showed sampling adequacy and sufficient inter-item correlations for PCA, respectively. The eigenvalue and scree plot test suggest that two components underpin the data, whereas the results of the parallel and Velicer's MAP test revealed one component. Given the robustness of the latter criteria [27], it was concluded that the psychosis-risk items constitute a unidimensional structure.

## Prevalence and demographic correlates of psychosis risk

As shown in Table 1, of the 589 participants who responded to the psychosis risk questionnaire, 54.2% ($n$ = 319) were classified as no/low risk for psychosis, 27.3% ($n$ = 161) as moderate risk and 18.5% ($n$ = 109) as high risk. In terms of the distress associated with PLEs, 44.3% reported they were not distressed, whereas 32.2% and 23.4% were a bit/quite and very distressed, respectively. Psychosis risk and distress experience were significantly correlated, ($\chi^2$ = 35.10, $p$ < 0.001), suggesting that participants in the high risk psychosis group were more likely to report that they were distressed by the PLEs. Psychosis risk was also significantly

**Table 1. Chi square results of the correlation between psychosis risk group and categorical variables.**

| Variables | Psychosis risk | | | Statistics | |
|---|---|---|---|---|---|
| | No/low | Moderate | High | Chi square | P-value |
| **Trimester** | | | | 14.18 | 0.007 |
| First | 26(8.2) | 19(11.8) | 170(9.2) | | |
| Second | 123(38.6) | 83(51.6) | 59(50.5) | | |
| Third | 170(53.3) | 59(36.6) | 44(40.4) | | |
| **Education** | | | | 39.42 | 0.000 |
| SHS/Equivalence | 133(43.5) | 98(62) | 62(60.2) | | |
| Diploma | 76(24.8) | 42(26.6) | 32(31.1) | | |
| Degree | 97(31.7) | 18(11.4) | 9(8.7) | | |
| **Partner Abuse** | | | | 29.85 | 0.000 |
| No | 234(74.8) | 85(53.5) | 56(52.3) | | |
| Yes | 79(25.2) | 74(46.5) | 51(47.7) | | |
| **Distress** | | | | 195.81 | 0.000 |
| No | 210(69.1) | 22(14.6) | 15(14.2) | | |
| A bit/quite | 74(24.3) | 65(43) | 37(34.9) | | |
| Very | 20(6.6) | 64(42.4) | 54(50.9) | | |
| **Self-Initiated Help-Seeking** | | | | 58.04 | 0.000 |
| No | 290(92.1) | 119(74.8) | 66(61.1) | | |
| Yes | 25(7.9) | 40(25.2) | 42(38.9) | | |
| **Being Asked** | | | | 34.47 | 0.000 |
| No | 143(45.4) | 39(24.4) | 21(19.4) | | |
| Yes | 172(54.6) | 121(75.6) | 87(80.6) | | |
| **Offered Support** | | | | 59.27 | 0.000 |
| No | 206(66.2) | 58(36.3) | 34(31.5) | | |
| Yes | 105(33.8) | 102(63.7) | 74(68.5) | | |

**Table 2. Intercorrelation and descriptive statistics of continuous study variables.**

| | 1 | 2 | 3 | 4 | 5 | 6 |
|---|---|---|---|---|---|---|
| **1.** Psychosis Risk | 1 | | | | | |
| **2.** COVID-19 Concerns | 0.39* | 1 | | | | |
| **3.** Depressive Symptoms | 0.36* | 0.38* | 1 | | | |
| **4.** Anxiety Symptoms | 0.38* | 0.41* | 0.71* | 1 | | |
| **5.** Suicidal Ideation | 0.32* | 0.05 | 0.21* | 0.19* | 1 | |
| **6.** Sleep Difficulty | 0.41* | 0.47* | 0.41* | 0.45* | .17* | 1 |
| M | 5.60 | 4.62 | 14.83 | 12.30 | 3.69 | 4.31 |
| SD | 3.98 | 2.23 | 4.46 | 4.42 | 1.36 | 1.64 |
| Minimum | 0 | 2 | 9 | 6 | 3 | 2 |
| Maximum | 16 | 8 | 36 | 28 | 12 | 8 |
| Cronbach's Alpha ($\alpha$) | .83 | .92 | .79 | .85 | .71 | .86 |

* Correlation is significant at the 0.01 level (2-tailed).

associated with pregnancy trimester ($\chi^2 = 14.18$, $p = 0.007$), level of education ($\chi^2 = 39.42$, $p < 0.001$) experience of partner abuse ($\chi^2 = 29.85$, $p < 0.001$) and help-seeking for mental health ($p < 0.001$).

## MANOVA results of group differences on dependent variables

The intercorrelations between the study variables were summarized in Table 2. With respect to Table 1, except for suicidal ideation and COVID-19 concerns, the study variables were significantly and positively correlated ($p < 0.01$). The results from the MANOVA revealed that the multivariate effect of psychosis risk group membership was statistically significant, Wilk's lambda = .700, $F (10, 1020) = 19.86$, $p < 0.001$, $\eta^2 = .16$. The univariate F ratios and eta squared values together with the means and standard deviations of the groups for each dependent variable are shown in Table 3. The eta squared values ranges from 0.10 to 0.15. The groups differ significantly on all the dependent variables ($p < .001$). A Bonferroni post hoc test, applied to the dependent variables showed that the moderate and high-risk psychosis groups reported significantly greater COVID-19 concerns, depressive and anxiety symptoms, suicidal ideation and sleep difficulty than the No/Low risk group. The moderate and high-risk groups differ significantly only on sleep difficulty, with the high-risk psychosis group reporting more sleep difficulty than the moderate risk group.

**Table 3. ANOVA (F) ratios, means and standard deviations of the no/low, moderate and high risk groups on the study variables.**

| | | | No/Low risk ($n = 280$) | | Moderate ($n = 140$) | | High risk ($n = 96$) | |
|---|---|---|---|---|---|---|---|---|
| Variables | F(1, 513) | $\eta^2$ | M | SD | M | SD | M | SD |
| COVID-19 Concerns | 45.55* | 0.15 | 4.00 | 2.00 | 5.48 | 2.28 | 6.07 | 2.10 |
| Depressive Symptoms | 43.02* | 0.14 | 13.30 | 3.58 | 16.80 | 4.44 | 16.35 | 4.72 |
| Anxiety symptoms | 35.36* | 0.12 | 10.95 | 3.90 | 13.77 | 4.42 | 14.31 | 4.27 |
| Suicidal Ideation | 28.08* | 0.10 | 3.25 | .82 | 3.99 | 1.67 | 4.17 | 1.56 |
| Sleep Difficulty | 44.10* | 0.15 | 3.8 | 1.38 | 4.75 | 1.67 | 5.40 | 1.79 |

* = $p < .001$; $\eta^2$ = partial eta squared; M = Mean and SD = Standard deviations.

**Table 4. Predictors' unique contributions in the multinomial logistic regression (n = 516).**

| Predictor | $\chi^2$-test | df | p |
|---|---|---|---|
| COVID-19 Concerns | 23.06 | 2 | < 0.001 |
| Depressive Symptoms | 13.02 | 2 | 0.001 |
| Anxiety symptoms | 1.08 | 2 | .581 |
| Suicidal Ideation | 33.89 | 2 | < 0.001 |
| Sleep Difficulty | 10.32 | 2 | 0.006 |

## Predicting psychosis risk group membership

The results of the multinomial logistic regression revealed that the model containing the predictors was significantly different from the intercept-only model, $\chi^2$ (10, $n = 516$) = 175.96, $p < 0.001$. Using the deviance criterion, the model provided a good fit to the data, $\chi^2$ (858, $n = 516$) = 782. 24, $p = .969$. The predictors explained 33.3% of the variance in the psychosis-risk group membership (Nagelkerke $R^2$ = .33). Prediction success for group membership was modest, with an overall rate of 60.5%, and correct prediction rates of 88.2%, 29.3% and 25% for no/low, moderate and high-risk groups, respectively. The unique contribution of the predictors in the multinomial logistic regression is summarized in Table 4. All but anxiety symptoms independently contributed to the significance of the regression model (all $ps < .05$). The result evaluating the influence of the predictors (Table 5) showed that COVID-19 concerns, depressive symptoms and suicidal ideation were significant predictors of moderate risk for psychosis. More precisely, participants who expressed concerns relating to COVID-19, scored high on depressive symptoms and suicidal ideation were 1.60 (CI = 1.236 − 2.076), 1.82 (CI = 1.295 − 2.549) and 1.93 (CI = 1.453 − 2.566) times more likely to be in the moderate risk for psychosis group, respectively.

Participants who expressed suicidal ideation and concerns relating to COVID-19 were 1.96 (CI = 1.437 − 2.659) and 2.16 (CI = 1.598 − 2.928) times more likely to be classified into high risk for psychosis group, respectively. In this model, sleep difficulty also emerged as a

**Table 5. Parameter estimates contrasting the no/low risk psychosis group versus moderate and high risk groups (n = 516).**

| Model | B | SE-B | Wald | Ex(B) | 95% CI for Ex(B) |
|---|---|---|---|---|---|
| **Moderate Risk** | | | | | |
| Intercept | | | | | |
| COVID-19 Concerns | 0.47 | 0.12 | 28.76** | 1.60 | 1.236 − 2.076 |
| Depressive Symptoms | 0.60 | 0.13 | 12.69* | 1.82 | 1.295 − 2.549 |
| Anxiety Symptoms | 0.01 | 0.17 | 0.01 | 1.00 | .726 − 1.398 |
| Suicidal Ideation | 0.66 | 0.15 | 20.57** | 1.93 | 1.453 − 2.566 |
| Sleep Difficulty | 0.18 | 0.14 | 1.61 | 1.20 | .908 − 1.573 |
| **High Risk** | | | | | |
| Intercept | | | | | |
| COVID-19 Concerns | 0.67 | 0.16 | 18.22** | 1.95 | 1.437 − 2.659 |
| Depressive Symptoms | 0.24 | 0.20 | 1.41 | 1.27 | 0.858 −1.872 |
| Anxiety Symptoms | 0.18 | 0.19 | 0.90 | 1.20 | 0.823 − 1.748 |
| Suicidal Ideation | 0.77 | 0.15 | 24.95** | 2.16 | 1.598 −2.928 |
| Sleep Difficulty | 0.50 | 0.16 | 10.03* | 1.65 | 1.209 − 2.238 |

Note: The dependent variable was psychosis risk groups with no/low psychosis risk group as the reference category.

Degree of freedom (df) = 1; Ex(B) = odd ratios; SE-B = Standard error of B.

* = p < .01

** = p < .001.

statistically significant predictor; participants who reported sleep difficulty were 1.65 (CI = 1.209 – 2.238) more likely to be in the high risk for psychosis group. From the foregoing, the most consistent predictors of moderate and high-risk group membership were COVID-19 concerns and suicidal ideation.

## Discussions

Understanding psychosis risk in pregnancy will improve decision making in maternal mental health care. In this study, 18.5% of the participants were classified as high risk for psychosis, whereas 27.3% fell into the moderate risk group. Although data-driven, the validity of the psychosis risk group is partly proven by the ANOVA and post-hoc analyses results that showed that participants in the high or moderate group significantly endorsed the risk factors of poor perinatal mental health than those in the low-risk group. The 18.5 to 27.3% psychosis risk found in this study is similar to the 27% psychosis risk prevalence rate recorded among pregnant women in Peru [7].

Psychosis risk is notably elevated among pregnant women who were more concerned about the COVID-19 effects as well as expressed suicidal ideation. That is, the above risk factors separated pregnant women classified as low/no risk from those categorized as moderate and high risk for psychosis. The findings relating to COVID-19 and elevated scores on psychosis risk is in tandem with the existing literature that suggest that COVID-19 significantly affects the mental health and wellbeing of pregnant and postpartum women. Reviews conducted on COVID-19 and perinatal mental health have found that pregnant and postpartum women endorsed more symptoms of anxiety and depression during the COVID-19 pandemic compared with the previous non-pandemic times [28–30]. While the exact mechanism underlying this finding has not been explored in this study, it is posited that the decrease in access to mental health support services such as professionals and social networks occasioned by the COVID-19 mitigation measures could be implicated [28]. The enforcement of COVID-19 mitigation measures, notably lockdowns resulted in limited behavioral practices such as physical activity or exercises that serve as protective factors against mental health problems.

The connection between psychosis risk and suicidal ideation has long been established in related literature among non-pregnant and postpartum women [31–33]. Several mechanisms of action have been proposed, including the view that command hallucination in psychosis promotes suicidal ideations. Others have also maintained that individuals at risk for psychosis tend to manifest severe and multi-comorbid psychopathologies such as depression and anxiety [31]. The cumulative effect of the psychopathologies may increase the risk for suicidal tendencies. The findings reported in this study further attest to the robustness of the relationship between psychosis risk and suicidal tendencies, providing additional layer of evidence on the need to pay critical attention to psychosis risk in pregnancy.

Depressive tendencies and sleep difficulties are among the most widely reported mental health problems experienced by pregnant women [34, 35]. In non-pregnant population, studies have established a relationship between psychosis risk and depression symptoms [36]. The current study has extended the existing literature by demonstrating a positive association between psychosis risk, depressive symptoms and sleep difficulties. Depressive symptoms and sleep difficulties were significantly endorsed by participants who were classified as moderate and severe risk for psychosis, compared with those at low risk. Depressive symptoms have been identified as forming an essential component of the prodrome of schizophrenia [36]. By extension, depressive symptoms can be expressed by a person at high risk for psychosis. This is partly because the distress and other behavioral changes associated with the experiences of psychotic like symptoms such as hallucination and delusions can promote depressive feelings.

## Limitations

The study findings should be evaluated considering the following limitations. The cross-sectional design adopted in this study does not permit commentaries on causal relationship. The recruitment of participants with at least senior high school education for the study implies that the findings may not be applicable to those with less formal education. The findings could have been different with equal or similar number of pregnant women across the three trimesters. The use of convenience sampling increases the risk of selection bias. Lastly, although single or few-item measures are capable of representing complex variables, multiple-item questionnaires are known to have superior psychometric properties, which could have influenced the findings reported here [37, 38].

## Conclusions

The study has provided initial evidence regarding the prevalence of PLEs in pregnancy in a Western Africa country, Ghana. It is envisaged that the findings will serve as a wake-up call for researchers to investigate PLEs and psychosis in the perinatal period as a public health issue. Healthcare professionals should equally consider to include PLEs screening measures into the existing gamut of mental health screening tools used in the perinatal period. Once screened, education and awareness creation on psychosis and psychosis-risk should be undertaken to increase the knowledge-base of pregnant women. Information on the characteristics, manifestation of psychotic-like symptoms, the transition to full-blown psychotic disorders and negative impacts of psychosis on pregnancy and pregnancy outcomes should be incorporated into the education and awareness program. It is recommended that future studies are conducted to unearth the nature or mechanisms as well as the risk and protective factors of the trajectory of PLEs to clinically diagnosable psychosis.

## Supporting information

**S1 Dataset.**
(SAV)

## Author Contributions

**Conceptualization:** Samuel Adjorlolo.

**Data curation:** Samuel Adjorlolo, Caroline Dinam Badzi.

**Formal analysis:** Samuel Adjorlolo.

**Funding acquisition:** Samuel Adjorlolo.

**Methodology:** Samuel Adjorlolo, Gwendolyn Mensah, Caroline Dinam Badzi.

**Project administration:** Samuel Adjorlolo.

**Writing – original draft:** Samuel Adjorlolo.

**Writing – review & editing:** Samuel Adjorlolo, Gwendolyn Mensah, Caroline Dinam Badzi.

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
