## [Decision Letter · Decision Letter 0]

16 Aug 2021

PONE-D-21-20182

Psychosis Risk among Pregnant Women in Ghana

PLOS ONE

Dear Dr. Adjorlolo,

Thank you for submitting your manuscript to PLOS ONE. After careful consideration, we feel that it has merit but does not fully meet PLOS ONE’s publication criteria as it currently stands. Therefore, we invite you to submit a revised version of the manuscript that addresses the points raised during the review process.

We look forward to receiving your revised manuscript.

Kind regards,

Frank T. Spradley

Academic Editor

PLOS ONE

2. We note that you have stated that you will provide repository information for your data at acceptance. Should your manuscript be accepted for publication, we will hold it until you provide the relevant accession numbers or DOIs necessary to access your data. If you wish to make changes to your Data Availability statement, please describe these changes in your cover letter and we will update your Data Availability statement to reflect the information you provide."

3. Please include a copy of Table xxxx which you refer to in your text on page 11.

Reviewers' comments:

Reviewer's Responses to Questions

**Comments to the Author**

1. Is the manuscript technically sound, and do the data support the conclusions?

Reviewer #1: Yes

Reviewer #2: Partly

2. Has the statistical analysis been performed appropriately and rigorously? 

Reviewer #1: Yes

Reviewer #2: No

3. Have the authors made all data underlying the findings in their manuscript fully available?

Reviewer #1: No

Reviewer #2: Yes

4. Is the manuscript presented in an intelligible fashion and written in standard English?

Reviewer #1: Yes

Reviewer #2: Yes

5. Review Comments to the Author

Reviewer #1: Dear authors, this is an interesting article about mental health of pregnant women. I quote below my suggestions to your manuscript.

a. Specify the exact number of the original sample and that of the final sample. It is not understood

b. The duration of the study is not mentioned in the methodology

c. How was the sample approached? Was it done during the standard check-up or during hospitalization?

d. The week of pregnancy is not specified

e. In the Introduction section describe what are the (PLEs) symptoms. In addition, for which mental disorders these symptoms are risk factors

f. Specify what the (PLEs) symptoms include in the results (eg what the anxiety symptoms include)

g. The psychometric scales are not presented in a table

h. The conclusions are not supported by the results

i.Please reconsider your definition(s) in the article

Example: Psychotic-like experiences (PLEs) are subtle, subclinical hallucinations and delusions which are quite common in general population*

Please reconsider your conclusion in the article

Example: The risk of developing a severe mental illness in pregnancy is estimated to be 7.1 in 10,000 per year. New-onset acute psychosis during pregnancy is extremely rare**

*Remberk B. Clinical significance of psychotic-like experiences in children and adolescents. Psychiatr Pol 2017, 51:271-82.

**Watkins ME, Newport J. Psychosis in Pregnancy. Obstetrics & Gynecology 2009, 113: 1349.

Reviewer #2: This paper uses a cross-sectional design to investigate risk factors during pregnancy for postpartum psychosis (PPP). It adds to an important area of the perinatal mental health literature, as PPP is the most severe mental illness in the perinatal period and not many studies have investigated psychotic-like experiences during pregnancy as a risk factor for PPP. A strength of the study is the large sample size the authors obtained data from.

I have divided my comments into major and minor points.

Major:

1. This study feels incomplete without following up with women in the postpartum period to understand who went on to develop PPP within the sample. Given that the study examined risk factors/correlates of psychotic-like experiences in pregnancy, and PLEs can be a starting point of PPP, it would be helpful to know whether those women who scored high on the PQ-16 questionnaire and had identified correlates (depression, suicidal ideation, COVID-19 anxiety, sleep disturbances) actually developed PPP symptoms in the end. This way we can understand whether 1) the identified correlates may in themselves be risk factors for PPP, and 2) the PQ-16 questionnaire is a valid predictor of PPP in itself. Furthermore, it would help the authors to determine whether a cut-off score can be used on the PQ-16 to identify pregnant women at greatest risk.

2. Please discuss why the authors chose not to include participants with a history of mental health disorder/current mental health disorder as these are one of the biggest risk factors for PPP? Especially given that it was found that main correlates of high PLE scores were depressive and anxious symptoms and suicidal ideation, it would be important to understand whether pregnant women with histories of (or current) mental illness score highly on the PQ-16.

3. It would be beneficial to analyse more variables in the study variables list, such as socioeconomic status, social support, marital status, perceived stress, history of childhood maltreatment, i.e. further risk factors for PLEs and PPP.

4. With regard to the questions on suicidal ideation and intimate partner violence, please state whether a factor analysis was completed to determine that these items are valid extractions of an entire questionnaire.

5. Given that there was multicollinearity between many of the variables in Table 2, they should not have all been entered into a regression together.

Minor:

1. In the introduction, please discuss PPP prevalence across more societies/cultures than only the US and Sweden, as both are high-income countries.

2. Please discuss whether the PQ-16 has been previously validated as a predictor of PPP in a perinatal sample, rather than a predictor of psychosis in a general sample.

3. Related to point 2, please discuss whether the PQ-16 has been validated across various cultures and ethnicities.

4. Please reference Hazelgrove et al. (2021)’s paper on risk factors for postpartum psychosis.

5. Table 1: please indicate where the group-differences are in the chi-square and please indicate whether the values are listed as number (%) or %(number).

6. In addition to asking women whether they were concerned about COVID-19, were they also asked whether they had contracted the virus? This is especially important as COVID-19 has been found to cause psychotic-like symptoms in some patients.

6. PLOS authors have the option to publish the peer review history of their article (what does this mean?). If published, this will include your full peer review and any attached files.

Reviewer #1: No

Reviewer #2: No

---

## [Author Response · Author response to Decision Letter 0]

22 Oct 2021

Psychosis Risk among Pregnant Women in Ghana

PONE-D-21-20182

RESPONSE FROM AUTHORS: We have noted our responses to the reviewers’ comments below in yellow highlight. We appreciate the feedback from the reviewers.

Reviewer 1 Comments

Dear authors, this is an interesting article about mental health of pregnant women. I quote below my suggestions to your manuscript. 

a. Specify the exact number of the original sample and that of the final sample. It is not understood.

Authors’ response: A total of 702 women were recruited for the study but some were missing data on some study variables. Excluding participants with missing data in the analyses caused a reduction in the sample size across.

b. The duration of the study is not mentioned in the methodology

Authors’ response: This has been specified as follows: “Data were collected from September to October 2020”

c. How was the sample approached? Was it done during the standard check-up or during hospitalization?

Authors’ response: We have included this information in the revised manuscript as follows: “The participants, who often congregate at the antenatal clinics of the facilities for their antenatal services, were informed about the purpose and duration of the study, their responsibilities for participating in the study, ethical issues such as confidentiality, consent, anonymity and benefits of participation.” (page 6).

d. The week of pregnancy is not specified

Authors’ response: We included data on the weeks of pregnancy as follows: “Pregnant women in their first (≤ 12 weeks of pregnancy), second (≤ 24 weeks of pregnancy) or third trimesters (≤ 36 weeks of pregnancy) were recruited for the study.” (page 6).

e. In the Introduction section describe what are the (PLEs) symptoms. In addition, for which mental disorders these symptoms are risk factors 

Authors’ response: A brief description of PLEs is provided. The section now reads:

“PLEs are very common in the general population, appearing first in adolescence and sometimes in childhood (Zavos et al., 2014) and can be categorized as positive (e.g., perceptual abnormalities, delusional thoughts) or negative (e.g., social withdrawal, avolition) (Yung et al., 2009). Among the common examples of the PLEs are hearing voices, seeing things, and smelling things that other people do not hear, see or smell, respectively.” (page 4).

f. The psychometric scales are not presented in a table

Authors’ response: In table 2, we provided data on the Cronbach alpha for all the scales used in the study.

g. The conclusions are not supported by the results

Authors’ response: We have revised the conclusion section of the manuscript accordingly.

Please reconsider your definition(s) in the article

Example: Psychotic-like experiences (PLEs) are subtle, subclinical hallucinations and delusions which are quite common in general population* 

Authors’ response: We have expanded on the initial definition provided in the manuscript. The section now reads:

“PLEs are subclinical symptoms of psychosis that do not meet the threshold for clinical diagnosis as psychotic illness (Fusar-Poli et al., 2012; Hielscher et al., 2018). PLEs are very common in the general population, appearing first in adolescence and sometimes in childhood (Zavos et al., 2014) and can be categorized as positive (e.g., perceptual abnormalities, delusional thoughts) or negative (e.g., social withdrawal, avolition) (Yung et al., 2009). Among the common examples of the PLEs are hearing voices, seeing things, and smelling things that other people do not hear, see or smell, respectively.” (page 4).

*Remberk B. Clinical significance of psychotic-like experiences in children and adolescents. Psychiatr Pol 2017, 51:271-82. 

 **Watkins ME, Newport J. Psychosis in Pregnancy. Obstetrics & Gynecology 2009, 113: 1349. 

Authors’ response: Thank you very much for these articles. They are extremely helpful. 

Reviewer #2: This paper uses a cross-sectional design to investigate risk factors during pregnancy for postpartum psychosis (PPP). It adds to an important area of the perinatal mental health literature, as PPP is the most severe mental illness in the perinatal period and not many studies have investigated psychotic-like experiences during pregnancy as a risk factor for PPP. A strength of the study is the large sample size the authors obtained data from.

I have divided my comments into major and minor points.

Major:

1. This study feels incomplete without following up with women in the postpartum period to understand who went on to develop PPP within the sample. Given that the study examined risk factors/correlates of psychotic-like experiences in pregnancy, and PLEs can be a starting point of PPP, it would be helpful to know whether those women who scored high on the PQ-16 questionnaire and had identified correlates (depression, suicidal ideation, COVID-19 anxiety, sleep disturbances) actually developed PPP symptoms in the end. This way we can understand whether 1) the identified correlates may in themselves be risk factors for PPP, and 2) the PQ-16 questionnaire is a valid predictor of PPP in itself. Furthermore, it would help the authors to determine whether a cut-off score can be used on the PQ-16 to identify pregnant women at greatest risk.

Authors’ response: It is true that following up on the participants after delivery to determine those diagnosed with psychosis would add significantly to the findings of the study. Our main goal in this study was to determine the prevalence of psychotic like experiences (PLEs). Individuals scoring high on PLEs are at risk for psychotic illness and not necessarily that they would develop psychotic illness. Therefore, it is possible that none of the participants would have been diagnosed with psychotic illness at follow-up. The discussion would have been different if we had screened for psychosis as a mental disorder. In view of this, we are happy to contribute to the emerging literature on the prevalence and correlates of PLEs among pregnant women in Ghana. We are optimistic that the findings will serve the research community in terms of opening up future research areas. 

2. Please discuss why the authors chose not to include participants with a history of mental health disorder/current mental health disorder as these are one of the biggest risk factors for PPP? Especially given that it was found that main correlates of high PLE scores were depressive and anxious symptoms and suicidal ideation, it would be important to understand whether pregnant women with histories of (or current) mental illness score highly on the PQ-16.

Authors’ response: We initially collected data on individuals with a history of mental disorders (n = 5). However, our initial analyses reveal that they scored very high on the measures of anxiety and depression etc, emerging as outliers. To maintain sample purity, we agreed to exclude them. If their numbers were high, we could have compare them with those without diagnoses of mental disorders. 

3. With regard to the questions on suicidal ideation and intimate partner violence, please state whether a factor analysis was completed to determine that these items are valid extractions of an entire questionnaire.

Authors response: Thank you for the observation. Yes, we subjected the items measuring suicidal ideation and IPV to principal component analysis. We requested for one component given the small number of items. All the items loaded satisfactory on the component, with correlation coefficients ranging from .52 to.89. Based on PCA, we treated the added the items measuring suicidal ideation and IPV to obtain unidimensional scores that were used for the analyses 

4. Given that there was multicollinearity between many of the variables in Table 2, they should not have all been entered into a regression together.

Authors’ response: We observed a correlation of .71 between anxiety and depressive symptoms. Other correlations ranged from .19 to 41, which are normal for regression analysis. The correlation coefficient of .71 appear high but there was no evidence of multicollinearity in the regression analysis. 

Minor:

1. In the introduction, please discuss PPP prevalence across more societies/cultures than only the US and Sweden, as both are high-income countries.

Authors’ response: Thank you for the observation. Unfortunately, data is limited from other context. In a South African we have referenced in the revised manuscript, there was no information on prevalence. Rather, data were collected from individuals diagnosed with PPP. We have noted this in the revised manuscript (page 3).

2. Please discuss whether the PQ-16 has been previously validated as a predictor of PPP in a perinatal sample, rather than a predictor of psychosis in a general sample.

Authors’ response: The PQ-16 has been validated to screen for psychotic-like experiences in the general population (Ising et al., 2012) and pregnant women (Levey et al., 2018).

3. Related to point 2, please discuss whether the PQ-16 has been validated across various cultures and ethnicities.

Authors’ response: With respect to pregnant women, the PQ-16 has been validated in Peruvian pregnant women (Levey et al., 2018) only. 

4. Please reference Hazelgrove et al. (2021)’s paper on risk factors for postpartum psychosis.

Authors’ response: Thank you very much. The reference has been included in the revised manuscript. 

5. Table 1: Please indicate whether the values are listed as number (%) or %(number).

Authors’ response: We have indicated in the revised manuscript that the figures in the bracket represent percentages.

6. In addition to asking women whether they were concerned about COVID-19, were they also asked whether they had contracted the virus? This is especially important as COVID-19 has been found to cause psychotic-like symptoms in some patients.

Authors’ response: We did not ask the participants whether they contracted the virus. This is an excellent observation that would be incorporated into our future work.

---

## [Decision Letter · Decision Letter 1]

16 Nov 2021

PONE-D-21-20182R1Psychosis Risk among Pregnant Women in GhanaPLOS ONE

Dear Dr. Adjorlolo,

Thank you for submitting your manuscript to PLOS ONE. After careful consideration, we feel that it has merit but does not fully meet PLOS ONE’s publication criteria as it currently stands. Therefore, we invite you to submit a revised version of the manuscript that addresses the points raised during the review process.

We look forward to receiving your revised manuscript.

Kind regards,

Frank T. Spradley

Academic Editor

PLOS ONE

Reviewers' comments:

Reviewer's Responses to Questions

**Comments to the Author**

1. If the authors have adequately addressed your comments raised in a previous round of review and you feel that this manuscript is now acceptable for publication, you may indicate that here to bypass the “Comments to the Author” section, enter your conflict of interest statement in the “Confidential to Editor” section, and submit your "Accept" recommendation.

Reviewer #1: All comments have been addressed

2. Is the manuscript technically sound, and do the data support the conclusions?

Reviewer #1: Yes

3. Has the statistical analysis been performed appropriately and rigorously? 

Reviewer #1: Yes

4. Have the authors made all data underlying the findings in their manuscript fully available?

Reviewer #1: Yes

5. Is the manuscript presented in an intelligible fashion and written in standard English?

Reviewer #1: Yes

6. Review Comments to the Author

Reviewer #1: Dear Author, I quote below some points about your article

• Abstract: Perinatal psychosis is a very rare and not a common illness

• Introduction: It is important to mention at this point the risk factors for developing psychosis, such as bipolar disorder, family history, atomic history of psychotic episode etc.

• It would be better to summarize the psychometric tools

• Also, the history of the woman's mental illness has not been studied, social support or not, if pregnancy was desired, if there was pregnancy pathology…

• As for Covid, is there any information on whether women were exposed to the virus or not?

• In the conclusions, what interventions do you suggest to perinatal health care professionals regarding the early recognition of psychotic symptoms? What do you suggest to improve this situation?

• What measures should be taken to prevent further development of the disorder in postpartum period?

7. PLOS authors have the option to publish the peer review history of their article (what does this mean?). If published, this will include your full peer review and any attached files.

Reviewer #1: No

---

## [Author Response · Author response to Decision Letter 1]

20 Nov 2021

Psychosis Risk among Pregnant Women in Ghana

PONE-D-21-20182R1

RESPONSE FROM AUTHORS: We have noted our responses to the reviewers’ comments below in yellow highlight. We appreciate the feedback from the reviewers.

Reviewer Comment 1: Abstract: Perinatal psychosis is a very rare and not a common illness

Response: The statement has been revised as follows;

“Psychotic illness, although is rare, has been reported in the perinatal period.”

Reviewer Comment 2: Introduction: It is important to mention at this point the risk factors for developing psychosis, such as bipolar disorder, family history, atomic history of psychotic episode etc.

Authors’ response: Thank you. We have included in the manuscript risk factors culled from the perinatal factors as follows;

As a heterogeneous mental disorder, several risk factors have been documented, including childhood maltreatment (Hazelgrove et al., 2021), low socio-economic status, neighborhood level social deprivation (Radua et al., 2018). Among the perinatal factors implicated in psychosis include diabetes in pregnancy, antepartum haemorrhage, preeclampsia, maternal stress during pregnancy, pre-pregnancy and pregnancy maternal obesity and cord complications (Radua et al., 2018).

Reviewer Comment 3: It would be better to summarize the psychometric tools

Authors’ response: Thank you very much. We are not what you meant by summarize the psychometric tools. This study is not a validation study and so detailed psychometric properties was not presented. As much as possible, we presented the reliability of the questionnaires that are not categorical. This was done for each measure as far as practicable. 

Reviewer Comment 4: As for Covid, is there any information on whether women were exposed to the virus or not?

Authors’ response: Thank you. We are not aware of the exposure status of the women who took part in the study. We can only deduce that since they were not recruited from isolation centers, or their medical records did not contain evidence of COVID-19, they probably did not contract COVID-19. 

Reviewer Comment 5: In the conclusions, what interventions do you suggest to perinatal health care professionals regarding the early recognition of psychotic symptoms? What do you suggest to improve this situation?

Authors’ response: Thank you very much. We noted in the manuscript that this is one of the initial studies exploring psychotic-like-experiences in pregnancy. Therefore, we are careful to state the usefulness of the findings to healthcare delivery. At this stage, we can only recommend, as stated in the manuscript (conclusion section) that screening for psychotic-like experiences should be incorporated into the existing perinatal care services with the hope of picking at risk women for early intervention. 

Reviewer Comment 6: What measures should be taken to prevent further development of the disorder in postpartum period?

Response: This is a very important question. Unfortunately, we could not follow the participants from pregnancy to post-delivery periods to know the rate of conversion to full-blown psychosis. Therefore, we could not comment on measures to prevent the development of psychosis in the postpartum period.

---

## [Decision Letter · Decision Letter 2]

3 Dec 2021

PONE-D-21-20182R2Psychosis Risk among Pregnant Women in GhanaPLOS ONE

Dear Dr. Adjorlolo,

Thank you for submitting your manuscript to PLOS ONE. After careful consideration, we feel that it has merit but does not fully meet PLOS ONE’s publication criteria as it currently stands. Therefore, we invite you to submit a revised version of the manuscript that addresses the points raised during the review process.

We look forward to receiving your revised manuscript.

Kind regards,

Frank T. Spradley

Academic Editor

PLOS ONE

Journal Requirements:

Reviewers' comments:

Reviewer's Responses to Questions

**Comments to the Author**

1. If the authors have adequately addressed your comments raised in a previous round of review and you feel that this manuscript is now acceptable for publication, you may indicate that here to bypass the “Comments to the Author” section, enter your conflict of interest statement in the “Confidential to Editor” section, and submit your "Accept" recommendation.

Reviewer #1: All comments have been addressed

2. Is the manuscript technically sound, and do the data support the conclusions?

Reviewer #1: Partly

3. Has the statistical analysis been performed appropriately and rigorously? 

Reviewer #1: Yes

4. Have the authors made all data underlying the findings in their manuscript fully available?

Reviewer #1: Yes

5. Is the manuscript presented in an intelligible fashion and written in standard English?

Reviewer #1: Yes

6. Review Comments to the Author

Reviewer #1: Dear Authors,

You have made enough effort to improve the article. However, some difficult points remain unclear

• With regard to the scales of measurement, I referred to the possibility of presenting them in less detail

• Furthermore, even if a study is in its early stages clear conclusions from the study need to be presented. The conclusions also include the authors' suggestions as well as any interventions required

• In addition to identifying a significant mental health problem in the postpartum period, researchers should be able to come up with some preventative measures to reduce them. For example, identifying risk factors from pregnancy or early postpartum period, educating perinatal health care professionals on recognizing dangerous symptoms, educating the family on caring for women with mental disorders and others….

Regards

7. PLOS authors have the option to publish the peer review history of their article (what does this mean?). If published, this will include your full peer review and any attached files.

Reviewer #1: No

---

## [Author Response · Author response to Decision Letter 2]

8 Dec 2021

Reviewer #1: Dear Authors,

You have made enough effort to improve the article. However, some difficult points remain unclear

• With regard to the scales of measurement, I referred to the possibility of presenting them in less detail.

Authors’ response: Thank you very much for the feedback. It is our understanding that the scales of measurement have been presented in a manner that the research community can follow. We would be happy if this is accepted the way it is.

• Furthermore, even if a study is in its early stages clear conclusions from the study need to be presented. The conclusions also include the authors' suggestions as well as any interventions required.

• In addition to identifying a significant mental health problem in the postpartum period, researchers should be able to come up with some preventative measures to reduce them. For example, identifying risk factors from pregnancy or early postpartum period, educating perinatal health care professionals on recognizing dangerous symptoms, educating the family on caring for women with mental disorders and others….

Regards

Authors response: We are grateful for the comments. In the earlier version of the manuscript, we accepted your recommendation and included a conclusion statement. In this revised version, we have expanded on this conclusion section as follows;

“The study has provided initial evidence regarding the prevalence of PLEs in pregnancy in a Western Africa country, Ghana. It is envisaged that the findings will serve as a wake-up call for researchers to investigate PLEs and psychosis in the perinatal period as a public health issue. Healthcare professionals should equally consider to include PLEs screening measures into the existing gamut of mental health screening tools used in the perinatal period. Once screened, education and awareness creation on psychosis and psychosis-risk should be undertaken to increase the knowledge-base of pregnant women. Information on the characteristics, manifestation of psychotic-like symptoms, the transition to full-blown psychotic disorders and negative impacts of psychosis on pregnancy and pregnancy outcomes should be incorporated into the education and awareness program. It is recommended that future studies are conducted to unearth the nature or mechanisms as well as the risk and protective factors of the trajectory of PLEs to clinically diagnosable psychosis.”

---

## [Decision Letter · Decision Letter 3]

11 Jan 2022

Psychosis Risk among Pregnant Women in Ghana

PONE-D-21-20182R3

Dear Dr. Adjorlolo,

We’re pleased to inform you that your manuscript has been judged scientifically suitable for publication and will be formally accepted for publication once it meets all outstanding technical requirements.

Kind regards,

Frank T. Spradley

Academic Editor

PLOS ONE

Reviewers' comments:

Reviewer's Responses to Questions

**Comments to the Author**

1. If the authors have adequately addressed your comments raised in a previous round of review and you feel that this manuscript is now acceptable for publication, you may indicate that here to bypass the “Comments to the Author” section, enter your conflict of interest statement in the “Confidential to Editor” section, and submit your "Accept" recommendation.

Reviewer #1: All comments have been addressed

2. Is the manuscript technically sound, and do the data support the conclusions?

Reviewer #1: Yes

3. Has the statistical analysis been performed appropriately and rigorously? 

Reviewer #1: Yes

4. Have the authors made all data underlying the findings in their manuscript fully available?

Reviewer #1: Yes

5. Is the manuscript presented in an intelligible fashion and written in standard English?

Reviewer #1: Yes

6. Review Comments to the Author

Reviewer #1: Dear Authors

Thank you very much for responding to my recommendations immediately.

I wish you a good luck!

7. PLOS authors have the option to publish the peer review history of their article (what does this mean?). If published, this will include your full peer review and any attached files.

Reviewer #1: No

---

## [Editor Report · Acceptance letter]

25 Jan 2022

PONE-D-21-20182R3 

Psychosis Risk among Pregnant Women in Ghana 

Dear Dr. Adjorlolo:

I'm pleased to inform you that your manuscript has been deemed suitable for publication in PLOS ONE. Congratulations! Your manuscript is now with our production department. 

Kind regards, 

on behalf of

Dr. PLOS Manuscript Reassignment 

Staff Editor

PLOS ONE